# Forgetful Bayes and myopic planning: Human learning and decision-making in a bandit setting

**Shunan Zhang**
Department of Cognitive Science
University of California, San Diego
La Jolla, CA 92093
s6zhang@ucsd.edu

**Angela J. Yu**
Department of Cognitive Science
University of California, San Diego
La Jolla, CA 92093
ajyu@ucsd.edu

## Abstract

How humans achieve long-term goals in an uncertain environment, via repeated trials and noisy observations, is an important problem in cognitive science. We investigate this behavior in the context of a multi-armed bandit task. We compare human behavior to a variety of models that vary in their representational and computational complexity. Our result shows that subjects' choices, on a trial-to-trial basis, are best captured by a "forgetful" Bayesian iterative learning model [21] in combination with a partially myopic decision policy known as Knowledge Gradient [7]. This model accounts for subjects' trial-by-trial choice better than a number of other previously proposed models, including optimal Bayesian learning and risk minimization, ε-greedy and win-stay-lose-shift. It has the added benefit of being closest in performance to the optimal Bayesian model than all the other heuristic models that have the same computational complexity (all are significantly less complex than the optimal model). These results constitute an advancement in the theoretical understanding of how humans negotiate the tension between exploration and exploitation in a noisy, imperfectly known environment.

## 1 Introduction

How humans achieve long-term goals in an uncertain environment, via repeated trials and noisy observations, is an important problem in cognitive science. The computational challenges consist of the learning component, whereby the observer updates his/her representation of knowledge and uncertainty based on ongoing observations, and the control component, whereby the observer chooses an action that balances between the short-term objective of acquiring reward and the long-term objective of gaining information about the environment. A classic task used to study such sequential decision making problems is the multi-arm bandit paradigm [15]. In a standard bandit setting, people are given a limited number of trials to choose among a set of alternatives, or arms. After each choice, an outcome is generated based on a hidden reward distribution specific to the arm chosen, and the objective is to maximize the total reward after all trials. The reward gained on each trial both has intrinsic value and informs the decision maker about the relative desirability of the arm, which can help with future decisions. In order to be successful, decision makers have to balance their decisions between general exploration (selecting an arm about which one is ignorant) and exploitation (selecting an arm that is known to have relatively high expected reward).

Because bandit problem elegantly capture the tension between exploration and exploitation that is manifest in real-world decision-making situations, they have received attention in many fields, including statistics [10], reinforcement learning [11, 19], economics [1, e.g.], psychology and neuroscience [5, 4, 18, 12, 6]. There is no known analytical optimal solution to the general bandit problem, though properties about the optimal solution of special cases are known [10]. For relatively simple, finite-horizon problems, the optimal solution can be computed numerically via dynamic program-

ming [11], though its computational complexity grows exponentially with the number of arms and trials. In the psychology literature, a number of heuristic policies, with varying levels of complexity in the learning and control processes, have been proposed as possible strategies used by human subjects [5, 4, 18, 12]. Most models assume that humans either adopt simplistic policies that retain little information about the past and sidestep long-term optimization (e.g. win-stay-lose-shift and $\epsilon$-greedy), or switch between an exploration and exploitation mode either randomly [5] or discretely over time as more is learned about the environment [18].

In this work, we analyze a new model for human bandit choice behavior, whose learning component is based on the dynamic belief model (DBM) [21], and whose control component is based on the *knowledge gradient* (KG) algorithm [7]. DBM is a Bayesian iterative inference model that assumes that there exists statistical patterns in a sequence of observations, and they tend to change at a characteristic timescale [21]. DBM was proposed as a normative learning framework that is able to capture the commonly observed *sequential effect* in human choice behavior, where choice probabilities (and response times) are sensitive to the local history of preceding events in a systematic manner — even if the subjects are instructed that the design is randomized, so that any local trends arise merely by chance and not truly predictive of upcoming stimuli [13, 8, 20, 3]. KG is a myopic approximation to the optimal policy for sequential informational control problem, originally developed for operations research applications [7]; KG is known to be exactly optimal in some special cases of bandit problems, such as when there are only two arms. Conditioned on the previous observations at each step, KG chooses the option that maximizes the future cumulative reward gain, based on the myopic assumption that the next observation is the last exploratory choice, and all remaining choices will be exploitative (choosing the option with the highest expected reward by the end of the next trial). Note that this myopic assumption is only used in reducing the complexity of computing the expected value of each option, and not actually implemented in practice – the algorithm may end up executing arbitrarily many non-exploitative choices. KG tends to explore more when the number of trials left is large, because finding an arm with even a slightly better reward rate than the currently best known one can lead to a large cumulative advantage in future gain; on the other hand, when the number of trials left is small, KG tends to stay with the currently best known option, as the relative benefit of finding a better option diminishes against the risk of wasting limited time on a good option. KG has been shown to outperform several established models, including the optimal Bayesian learning and risk minimization, $\epsilon$-greedy and win-stay-lose-shift, for human decision-making in bandit problems, under two certain learning scenarios other than DBM [22].

In the following, we first describe the experiment, then describe all the learning and control models that we consider. We then compare the performance of the models both in terms of agreement with human behavior on a trial-to-trial basis, and in terms of computational optimality.

## 2   Experiment

We adopt data from [18], where a total of 451 subjects participated in the experiment as part of "testweek" at the University of Amsterdam. In the experiment, each participant completed 20 bandit problems in sequence, all problems had 4 arms and 15 trials. The reward rates were fixed for all arms in each game, and were generated, prior to the start of data collection, independently from a $\text{Beta}(2,2)$ distribution. All participants played the *same* reward rates, but the order of the games was randomized. Participants were instructed that the reward rates in all games were drawn from the same environment, and that the reward rates were drawn only once; participants were not told the exact form of the Beta environment, i.e. $\text{Beta}(2,2)$. A screenshot of the experimental interface is shown in Fig 1:a.

## 3   Models

There exist multiple levels of complexity and optimality in both the learning and the decision components of decision making models of bandit problems. For the learning component, we examine whether people maintain any statistical representation of the environment at all, and if they do, whether they only keep a mean estimate (running average) of the reward probability of the different options, or also uncertainty about those estimates; in addition, we consider the possibility that they entertain trial-by-trial fluctuation of the reward probabilities. The decision component can also

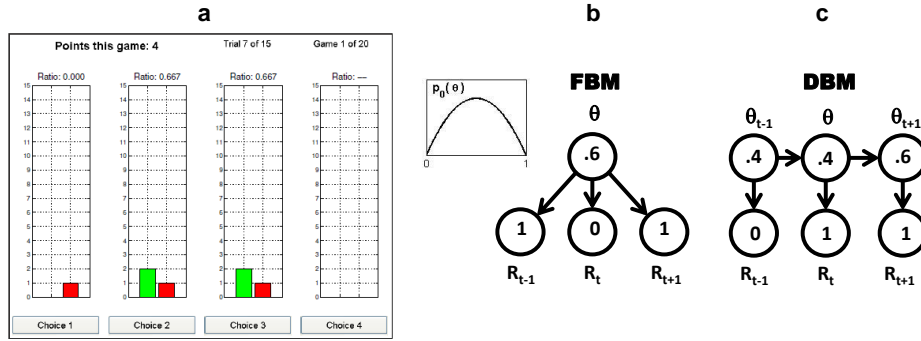

Figure 1: (a) A screenshot of the experimental interface. The four panels correspond to the four arms, each of which can be chosen by clicking the corresponding button. In each panel, successes from previous trials are shown as green bars, and failures as red bars. At the top of each panel, the ratio of successes to failures, if defined, is shown. The top of the interface provides the count of the total number of successes to the current trial, index of the current trial and index of the current game. (b) Bayesian graphical model of FBM, assuming fixed reward probabilities. $\theta \in [0,1]$, $R_t \in \{0,1\}$. The inset shows an example of the Beta prior for the reward probabilities. The numbers in circles show example values for the variables. (c) Bayesian graphical model of DBM, assuming reward probabilities change from trial to trial. $P(\theta_t) = \gamma\delta(\theta_t = \theta_{t-1}) + (1-\gamma)P_0(\theta_t)$.

differ in complexity in at least two respects: the objective the decision policy tries to optimize (e.g. reward versus information), and the time-horizon over which the decision policy optimizes its objective (e.g. greedy versus long-term). In this section, we introduce models that incorporate different combinations of learning and decision policies.

## 3.1 Bayesian Learning in Beta Environments

The observations are generated independently and identically (iid) from an unknown Bernoulli distribution for each arm. We consider two Bayesian learning scenarios below, the dynamic belief model (DBM), which assumes that the Bernoulli reward rates for all the arms can reset on any trial with probability $1-\gamma$, and the fixed belief model (FBM), a special case of DBM that assumes the reward rates to be stationary throughout each game. In either case, we assume the prior distribution that generates the Bernoulli rates is a Beta distribution, Beta$(\alpha, \beta)$, which is conjugate to the Bernoulli distribution, and whose two hyper-parameters, $\alpha$ and $\beta$, specify the pseudo-counts associated with the prior.

### 3.1.1 Dynamic Belief Model

Under the dynamic belief model (DBM), the reward probabilities can undergo discrete changes at times during the experimental session, such that at any trial, the subject's prior belief is a mixture of the posterior belief from the previous trial and a generic prior. The subject's implicit task is then to track the *evolving* reward probability of each arm over the course of the experiment.

Suppose on each game, we have $K$ arms with reward rates, $\theta_k, k = 1, \cdots, K$, which are iid generated from Beta$(\alpha, \beta)$. Let $S_k^t$ and $F_k^t$ be the numbers of successes and failures obtained from the $k$th arm on the trial $t$. The estimated reward probability of arm $k$ at trial $t$ is $\theta_k^t$. We assume $\theta_k^t$ has a Markovian dependence on $\theta_k^{t-1}$, such that there is a probability $\gamma$ of them being the same, and a probability $1-\gamma$ of $\theta_k^t$ being redrawn from the prior distribution Beta$(\alpha, \beta)$. The Bayesian ideal observer combines the sequentially developed prior belief about reward probabilities, with the incoming stream of observations (successes and failures on each arm), to infer the new posterior distributions. The observation $R_k^t$ is assumed to be Bernoulli, $R_k^t \sim$ Bernoulli $(\theta_k^t)$. We use the notation $q_k^t(\theta_k^t) := \text{Pr}(\theta_k^t | S_k^t, F_k^t)$ to denote the posterior distribution of $\theta_k^t$ given the observed sequence, also known as the *belief state*. On each trial, the new posterior distribution can be computed via Bayes' Rule:

$$q_k^t(\theta_k^t) \sim \text{Pr}(R_k^t | \theta_k^t) \text{Pr}(\theta_k^t | S_k^{t-1}, F_k^{t-1}) \tag{1}$$

where the prior probability is a weighted sum (parameterized by $\gamma$) of last trial's posterior and the generic prior $q^0 := \text{Beta}(\alpha, \beta)$:

$$\Pr\left(\theta_k^t = \theta | S_k^{t-1}, F_k^{t-1}\right) = \gamma q_k^{t-1}(\theta) + (1-\gamma)q^0(\theta) \tag{2}$$

### 3.1.2 Fixed Belief Model

A simpler generative model (and more correct one given the true, stationary environment) is to assume that the statistical contingencies in the task remain fixed throughout each game, i.e. all bandit arms have fixed probabilities of giving a reward throughout the game. What the subjects would then learn about the task over the time course of the experiment is the true value of $\theta$. We call this model a fixed belief model (FBM); it can be viewed as a special case of the DBM with $\gamma = 1$. In the Bayesian update rule, the prior on each trial is simply the posterior on the previous trial.

Figure 1b;c illustrates the graphical models of FBM and DBM, respectively.

## 3.2 Decision Policies

We consider four different decision policies. We first describe the optimal model, and then the three heuristic models with increasing levels of complexity.

### 3.2.1 The Optimal Model

The learning and decision problem for bandit problems can be viewed as as a Markov Decision Process with a finite horizon [11], with the state being the belief state $\mathbf{q}^t = (q_1^t, q_2^t, q_3^t, q_4^t)$, which obviously provides the sufficient statistics for all the data seen up through trial $t$. Due to the low dimensionality of the bandit problem here (i.e. small number of arms and number of trials per game), the optimal policy, up to a discretization of the belief state, can be computed numerically using Bellman's dynamic programming principle [2]. Let $V^t(\mathbf{q}^t)$ be the expected total future reward on trial $t$. The optimal policy should satisfy the following iterative property:

$$V^t(\mathbf{q}^t) = \max_k \theta_k^t + \mathbb{E}\left[V^{t+1}(\mathbf{q}^{t+1})\right] \tag{3}$$

and the optimal action, $D^t$, is chosen according to

$$D^t(\mathbf{q}^t) = \text{argmax}_k \theta_k^t + \mathbb{E}\left[V^{t+1}(\mathbf{q}^{t+1})\right] \tag{4}$$

We solve the equation using dynamic programming, backward in time from the last time step, whose value function and optimal policy are known for any belief state: always choose the arm with the highest expected reward, and the value function is just that expected reward. In the simulations, we compute the optimal policy off-line, for any conceivable setting of belief state on each trial (up to a fine discretization of the belief state space), and then apply the computed policy for each sequence of choice and observations that each subject experiences. We use the term "the optimal solution" to refer to the specific solution under $\alpha = 2$ and $\beta = 2$, which is the true experimental design.

### 3.2.2 Win-Stay-Lose-Shift

WSLS does not learn any abstract representation of the environment, and has a very simple decision policy. It assumes that the decision-maker will keep choosing the same arm as long as it continues to produce a reward, but shifts to other arms (with equal probabilities) following a failure to gain reward. It starts off on the first trial randomly (equal probability at all arms).

### 3.2.3 $\varepsilon$-Greedy

The $\varepsilon$-greedy model assumes that decision-making is determined by a parameter $\varepsilon$ that controls the balance between random exploration and exploitation. On each trial, with probability $\varepsilon$, the decision-maker chooses randomly (exploration), otherwise chooses the arm with the greatest estimated reward rate (exploitation). $\varepsilon$-Greedy keeps simple estimates of the reward rates, but does not track the uncertainty of the estimates. It is not sensitive to the horizon, maximizing the immediate gain with a constant rate, otherwise searching for information by random selection.

More concretely, ε-greedy adopts a stochastic policy:

$$\Pr\left(D^t = k \,|\, \varepsilon, \theta^t\right) = \begin{cases} (1-\varepsilon)/M^t & \text{if } k \in \operatorname{argmax}_{k'} \theta_{k'}^t \\ \varepsilon/(K-M^t) & \text{otherwise} \end{cases}$$

where $M^t$ is the number of arms with the greatest estimated value at the $t$th trial.

### 3.2.4 Knowledge Gradient

The knowledge gradient (KG) algorithm [16] is an approximation to the optimal policy, by *pretending* only one more exploratory measurement is allowed, and assuming all remaining choices will exploit what is known after the next measurement. It evaluates the expected change in each estimated reward rate, if a certain arm were to be chosen, based on the current belief state. Its approximate value function for choosing arm $k$ on trial $t$ given the current belief state $\mathbf{q}^t$ is

$$v_k^{\text{KG},t} = \mathbb{E}\left[\max_{k'} \theta_{k'}^{t+1} \,|\, D^t = k, \mathbf{q}^t\right] - \max_{k'} \theta_{k'}^t \tag{5}$$

The first term is the expected largest reward rate (the value of the subsequent exploitative choices) on the next step if the $k$th arm were to be chosen, with the expectation taken over all possible outcomes of choosing $k$; the second term is the expected largest reward given no more exploitative choices; their difference is the "knowledge gradient" of taking one more exploratory sample.

The KG decision rule is

$$D^{\text{KG},t} = \arg\max_k \theta_k^t + (T-t-1)\, v_k^{\text{KG},t} \tag{6}$$

The first term of Equation 6 denotes the expected immediate reward by choosing the $k$th arm on trial $t$, whereas the second term reflects the expected knowledge gain. The formula for calculating $v_k^{\text{KG},t}$ for the binary bandit problems can be found in Chapter 5 of [14].

### 3.3 Model Inference and Evaluation

Unlike previous modeling papers on human decision-making in the bandit setting [5, 4, 18, 12], which generally look at the average statistics of how people distribute their choices among the options, here we use a more stringent trial-by-trial measure of the *model agreement*, i.e. how well each model captures subject's choice. We calculate the per-trial likelihood of the subject's choice conditioned on the previously experienced actions and choices. For WSLS, it is 1 for a win-stay decision, $1/3$ for a lose-shift decision (because the model predicts shifting to the other three arms with equal probability), and 0 otherwise. For probabilistic models, take ε-greedy for example, it is $(1-\varepsilon)/M$ if the subject chooses the option with the highest predictive reward, where $M$ is the number of arms with the highest predictive reward; it is $\varepsilon/(4-M)$ for any other choice, and when $M = 4$, it is considered all arms have the highest predictive reward.

We use sampling to compute a posterior distribution of the following model parameters: the parameters of the prior Beta distribution ($\alpha$ and $\beta$) for all policies, $\gamma$ for all DBM policies, $\varepsilon$ for ε-greedy. For this model fitting process, we infer the re-parameterization of $\alpha/(\alpha+\beta)$ and $\alpha+\beta$, with a uniform prior on the former, and weakly informative prior for the latter, i.e. $\Pr(\alpha+\beta) \sim (\alpha+\beta)^{-3/2}$, as suggested by [9]. The reparameterization has psychological interpretation as the mean reward probability and the certainty. We use uniform prior for $\varepsilon$ and $\gamma$. Model inference use combined sampling algorithm, with Gibbs sampling of $\varepsilon$, and Metropolis sampling of $\gamma$, $\alpha$ and $\beta$. All chains contained 3000 steps, with a burn-in size of 1000. All chains converged according to the R-hat measure [9]. We calculate the average per-trial likelihood (across trials, games, and subjects) under each model based on its maximum *a posteriori* (MAP) parameterization.

We fit each model across all subjects, assuming that every subject shared the same prior belief of the environment ($\alpha$ and $\beta$), rate of exploration ($\varepsilon$), and rate of change ($\gamma$). For further analyses to be shown in the result section, we also fit the ε-greedy policy and the KG policy together with both learning models for each individual subject. All model inferences are based on a leave-one-out cross-validation containing 20 runs. Specifically, for each run, we train the model while withholding one game (sampled without replacement) from each subject, and test the model on the withheld game.

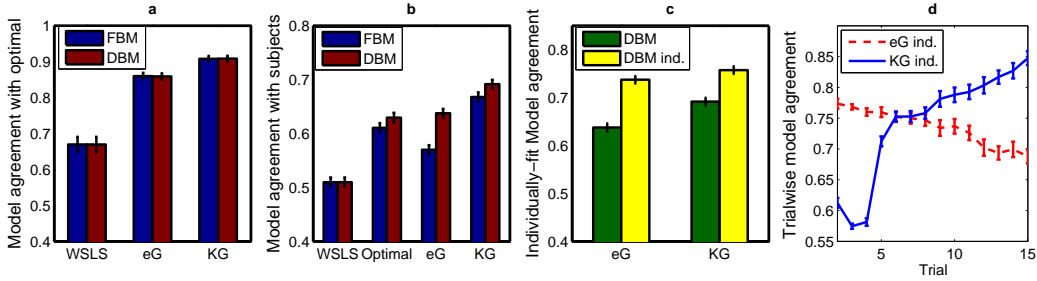

Figure 2: (a) Model agreement with data simulated by the optimal solution, measured as the average per-trial likelihood. All models (except the optimal) are fit to data simulated by the optimal solution under the correct beta prior Beta$(2,2)$. Each bar shows the mean per-trial likelihood (across all subjects, trials and games) of a decision policy coupled with a learning framework. For ε-greedy (eG) and KG, the error bars show the standard errors of the mean per-trial likelihood calculated across all tests in the cross validation procedure (20-fold). WSLS does not rely on any learning framework.(b) Model agreement with human data based on a leave-one(game)-out cross-validation, where we randomly withhold one game from each subject for training, i.e. we train the model on a total number of $19 \times 451$ games, with 19 games from each subject. For the current study, we implement the optimal policy under DBM using the estimated γ under the KG DBM model in order to reduce the computational burden. (c) Mean per-trial likelihood of the ε-greedy model (eG) and KG with individually-fit parameters (for each subject), using cross-validation; the individualized (ind. for abbreviation in the legend) DBM assumes each person has his/her own Beta prior and γ. (d) Trialwise agreement of eG and KG under individually-fit MAP parameterization. The mean per-trial likelihood is calculated across all subjects for each trial, with the error bars showing the standard error of the mean per-trial likelihood across all tests.

## 4 Results

### 4.1 Model agreement with the Optimal Policy

We first examine how well each of the decision policies agrees with the optimal policy on a trial-to-trial basis. Figure 2a shows the mean per-trial likelihood (averaged across all tests in the cross-validation procedure) of each model, when fit to data simulated by the optimal solution under the true design Beta(2,2). KG algorithm, under either learning framework, is most consistent (over 90%) with the optimal algorithm (separately under FBM and DBM assumptions). This is not surprising given that KG is an approximation algorithm to the optimal policy. The inferred prior is Beta$(1.93, 2.15)$, correctly recovering the actual environment. The simplest WSLS model, on the other hand, achieves model agreement well above 60%. In fact, the optimal model also almost always stays after a success; the only situation that WSLS does not resemble the optimal decision occurs when it shifts away from an arm that the optimal policy would otherwise stay with. Because the optimal solution (which simulated the data) knows the true environment, DBM does not have advantage against FBM.

### 4.2 Model Agreement with Human Data

Figure 2b shows the mean per-trial likelihood (averaged across all tests in the cross-validation procedure) of each model, when fit to the human data. KG with DBM outperforms other models of consideration. The average posterior mean of γ across all tests is .81, with standard error .091. The average posterior means for α and β are .65 and 1.05, with standard errors .074 and .122, respectively. A γ value of .81 implies that the subjects behave as if they think the world changes on average about every 5 steps (calculated as $1/(1 - .81)$).

We did a pairwise comparison between models on the mean per-trial likelihood of the subject's choice given each model's predictive distribution, using a pairwise t-test. The test between DBM-

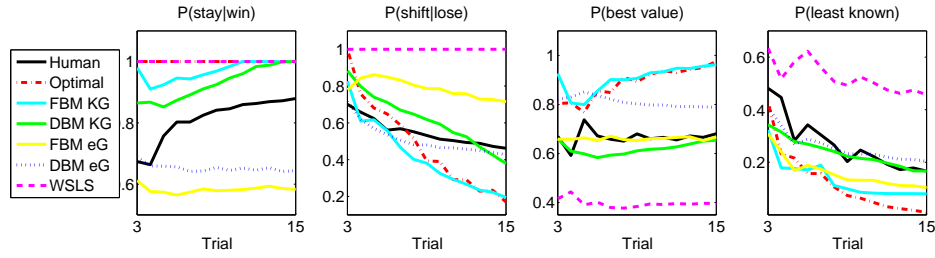

Figure 3: Behavioral patterns in the human data and the simulated data from all models. The four panels show the trial-wise probability of staying after winning, shifting after losing, choosing the greatest estimated value on any trial, choosing the least known when the exploitative choice is not chosen, respectively. Probabilities are calculated based on simulated data from each model at their MAP parameterization, and are averaged across all games and all participants. The optimal solution shown here uses the correct prior $\text{Beta}(2, 2)$.

optimal and DBM-eG, and the test between DBM-optimal and FBM-optimal, are not significant at the .05 level. All other tests are significant. Table 1 shows the p-values for each pairwise comparison.

Table 1: P-values for all pairwise t tests.

| KG DB | | | | | KG FB | | | | eG DB | | | eG FB | | Op DB |
|---|---|---|---|---|---|---|---|---|---|---|---|---|---|---|
| KG FB | eG DB | eG FB | Op DB | Op FB | eG DB | eG FB | Op DB | Op FB | eG FB | Op DB | Op FB | Op DB | Op FB | Op FB |
| .0480 | .0001 | .0000 | .0001 | .0000 | .0187 | .0000 | .0060 | .0002 | .0001 | .5066 | .0354 | .0001 | .0036 | .1476 |

Figure 2c shows the model agreement with human data, of ε-greedy and KG, when their parameters are individually fit. KG with DBM with individual parameterization has the best performance under cross validation. ε-Greedy also has a great gain in model agreement when coupled with DBM. In fact, under DBM, ε-greedy and KG have close performance in the overall model agreement. However, Figure 2d shows a systematic difference between the two models in their agreement with human data on a trial-by-trial base: during early trials, subjects' behavior is more consistent with ε-greedy, whereas during later trials, it is more consistent with KG.

We next break down the overall behavioral performance into four finer measures: how often people do win-stay and lose-shift, how often they exploit, and whether they use random selection or search for the greatest amount of information during exploration. Figure 3 shows the results of model comparisons on these additional behavioral criteria. We show the patterns of the subjects, the optimal solution with Beta(2,2), KG and eG under both learning frameworks and the simplest WSLS.

The first panel, for example, shows the trialwise probability of staying with the same arm following a previous success. People do not stay with the same arm after an immediate reward, which is always the case for the optimal algorithm. Subjects also do not persistently explore, as predicted by ε-greedy. In fact, subjects explore more during early trials, and become more exploitative later on, similar to KG. As implied by Equation 5, KG calculates the probability of an arm surpassing the known best upon chosen, and weights the knowledge gain more heavily in the early stage of the game. During the early trials, it sometimes chooses the second-best arm to maximize the knowledge gain. Under DBM, a previous success will cause the corresponding arm to appear more rewarding, resulting in a smaller knowledge gradient value; because knowledge is weighted more heavily during the early trials, the KG model then tends to choose the second best arms that have a larger knowledge gain.

The second panel shows the trialwise probability of shifting away given a previous failure. When the horizon is approaching, it becomes increasingly important to stay with the arm that is known to be reasonably good, even if it may occasionally yield a failure. All algorithms, except for the naive WSLS algorithm, show a downward trend to shift after losing as the horizon approaches, along with human choices. ε-Greedy with DBM learning is closest to human behavior.

The third panel shows the probability of choosing the arm with the largest success ratio. KG, under FBM, mimics the optimal model in that the probability of choosing the highest success ratio increases over time; they both grossly overly estimate subjects' tendency to select the highest success

ratio, as well as predicting an unrealized upward trend. WSLS under-estimates how often subjects make this choice, while ε-greedy under DBM learning over-estimates it. It is KG under DBM, and ε-greedy with FBM, that are closest to subjects' behavior.

The fourth panels shows how often subjects choose to explore the least known option when they shift away from the choice with the highest expected reward. It is DBM with either KG or ε-greedy that provides the best fit.

In general, the KG model with DBM matches the second-order trend of human data the best, with ε-greedy following closely behind. However, there still exists a gap on the absolute scale, especially with respect to the probability of staying with a successful arm.

# 5    Discussion

Our analysis suggests that human behavior in the multi-armed bandit task is best captured by a knowledge gradient decision policy supported by a dynamic belief model learning process. Human subjects tend to explore more often than policies that optimize the specific utility of the bandit problems, and KG with DBM attributes this tendency to the belief of a stochastically changing environment, causing the sequential effects due to recent trial history. Concretely, we find that people adopt a learning process that (erroneously) assumes the world to be non-stationary, and that they employ a semi-myopic choice policy that is sensitive to the horizon but assumes one-step exploration when comparing action values.

Our results indicate that all decision policies considered here capture human data much better under the dynamic belief model than the fixed belief model. By assuming the world is changeable, DBM discount data from the distant past in favor of new data. Instead of attributing this discounting behavior to biological limitations (e.g. memory loss), DBM explains it as the automatic engagement of mechanisms that are critical for adapting to a changing environment. Indeed, there is previous work suggesting that people approach bandit problems as if expecting a changing world [17]. This is despite informing the subjects that the arms have fixed reward probabilities.

So far, our results also favor the knowledge gradient policy as the best model for human decision-making in the bandit task. It optimizes the semi-myopic goal of maximizing future cumulative reward while assuming only one more time step of exploration and strict exploitation thereafter. The KG model under the more general DBM has the largest proportion of correct predictions of human data, and can capture the trial-wise dynamics of human behavioral reasonably well. This result implies that humans may use a normative way, as captured by KG, to explore by combining immediate reward expectation and long-term knowledge gain, compared to the previously proposed behavioral models that typically assumes that exploration is random or arbitrary. In addition, KG achieves similar behavioral patterns as the optimal model, and is computationally much less expensive (in particular being online and incurring a constant cost), making it a more plausible algorithm for human learning and decision-making.

We observed that decision policies vary systematically in their abilities to predict human behavior on different kinds of trials. In the real world, people might use hybrid policies to solve the bandit problems; they might also use some smart heuristics, which dynamically adjusts the weight of the knowledge gain to the immediate reward gain. Figure 2d suggests that subjects may be adopting a strategy that is aggressively greedy at the beginning of the game, and then switches to a policy that is both sensitive to the value of exploration and the impending horizon as the end of the game approaches. One possibility is that subjects discount future rewards, which would result in a more exploitative behavior than non-discounted KG, especially at the beginning of the game. These would all be interesting lines of future inquiries.

**Acknowledgments**

We thank M Steyvers and E-J Wagenmakers for sharing the data. This material is based upon work supported by, or in part by, the U. S. Army Research Laboratory and the U. S. Army Research Office under contract/grant number W911NF1110391 and NIH NIDA B/START # 1R03DA030440-01A1.

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
