[Supplementary Material · dbmKG_supp.pdf]

# Supplemental materials for "Forgetful Bayes and Myopic Planning: Human learning and decision-making in a bandit setting"

**Shunan Zhang**
Department of Cognitive Science
University of California, San Diego
La Jolla, CA 92093
s6zhang@ucsd.edu

**Angela J. Yu**
Department of Cognitive Science
University of California, San Diego
La Jolla, CA 92093
ajyu@ucsd.edu

## 1 Distribution of model agreement over individual subjects

We illustrate how well the models fit different individuals' decisions in Figure 1. We show the mean per-trial likelihood under each model, with the MAP estimate of the model parameters (fit across all subjects). The histogram shows that the model agreements for all models vary largely across individuals, suggesting large individual differences.

Figure 1: Model agreement distributions. x-axis: per-trial likelihood; y-axis: number of subjects