[Reviews · NeurIPS 2013]

Submitted by Assigned_Reviewer_4

This paper propose a novel model for human decision-making in the multi-arm bandit paradigm. The model combine Bayesian learning algorithms for sequence, a fixed belief model (FBM) and a dynamic belief model (DBM), and recently developed decision polity, that is, knowledge gradient. An interesting point is that the DBM which assumed the reward probability was dynamic gave the better fit to human data than FBM, although the subjects were informed that the reward probabilities were static.

My main concern is that the authors did not mention about the statistical significance for the model comparison results. The lack of information about what error bars in the Figure 2 represent further makes readers less confident about the differences in performance of models. Even the cross-validation procedure is used, some measure for statistical significance such as p-value in statistical hypothesis testing or Bayes factor is needed to consider the possibility that the resulting difference arose just by chance alone. See following paper for example:
Ito, M., & Doya, K. (2009). Validation of decision-making models and analysis of decision variables in the rat basal ganglia. Journal of Neuroscience, 29(31), 9861–74.

I think that the authors would better to include several other models for model comparison. For learning the reward probability, it could be a state space model (Kalman filter) which assumes that the reward probability continuously change. For action selection, it could be an ordinary softmax function. Although these are not necessary, it is beneficial for readers since previous influential studies like Daw et al., (Nature, 2006) examined such a model.
Summary: An interesting model and results. But the way the results were presented were not sufficiently convincing.

Submitted by Assigned_Reviewer_5

The work describes a comparison between human decision behavior in a decision task with uncertain reward structures with a set of models representing different decision strategies. The results suggest that a model that assumes a dynamic reward structure and employs a sub-optimal (myopic) decision strategy best accounts for human behavior in a multi-arm bandit task. Notably, an optimal model much less matches human behavior.

The work is of good quality. It involved an extended human behavioral experiment and a careful comparison of different models. The analyses are good and appropriate (e.g. leave-one-out) with the restriction that the authors do not address the different levels of complexities the model have in their comparison. For example, the DBM model clearly is more powerful than the model that assume as fixed reward structure and so it is not surprising that the DBM models (slightly) better fit the data.
One other aspect that deserves much more discussion is the fact that in the experiments subjects' i) knew that the reward structure is fixed, and ii) did not have to memorize any of their previous decisions nor the outcomes since the display kept track of those, yet the model that best accounted for the data was assuming a dynamic reward structure and some memory loss. This discrepancy is striking and it would be very interesting to know why.

The paper is clearly written and well presented.

The work is not extraordinarily original, given that it applies previously derived and suggested models to a particular data set. However, such work is needed and necessary, and thus valuable.

Significance is slightly diminished by the fact that it is not quite clear what to take away from the results.

Summary: The described work describes a worthy and clearly executed effort of comparing human data in a decision-making task to various models. The results are slightly difficult to interpret given obvious differences in the models' complexities and the fact that the overall model fit is not overly impressive; i.e. what do we take away from this work?

Submitted by Assigned_Reviewer_6

This paper seeks to understand human behavior in multi-armed bandit problems by comparing human performance to the predictions of four existing models. After presenting a comparison of the models’ predictions to human judgments, the authors conclude that the “knowledge gradient” (Gupta & Miescke, 1996; Frazier, Powell, & Dayanik, 2008) explains human behavior well if it assumes that reward distributions may change over time. The paper also includes an analysis of some trial-by-trial trends in the data and model predictions.

--Quality--

I didn't find any errors or conceptual problems in the model evaluation or the experimental design.

--Clarity--

The writing is quite clear in general, with some exceptions:

* On line 71 “Conditioned on the previous…” this sentence might lead a reader to think authors were claiming that KG is optimal in general.
* Section 3.3’s introduction made be think that the evaluation would be based directly on likelihoods of the human judgments under each model, and I had to re-read it after looking at the results.

--Originality--

To my knowledge, this is the first paper to compare multiple models against human behavior in multi-armed bandit problems, with the aim understanding the trade-off between exploration and exploitation. That said, the experimental design and the models -- aside from the DBM/FBM comparison -- are not new, and past bandit experiments have considered how humans approach the explore/exploit trade-off (e.g., Ejova, Navarro & Perfors, 2009).

--Significance--

My main worry about this paper is that it won't attract a wide audience at NIPS, because the conclusions aren't very far-reaching and it doesn't appear to have content that would draw an algorithms- or applications-oriented audience. Ultimately, the conclusion is that a particular model, KG, explains human behavior better than some plausible alternatives, and does better when it incorporates the assumption that bandits' rewards might change over time. It doesn't make a compelling case that the dynamic belief model is necessary in general, and there's past work that could be taken to suggest that people approach bandit problems as if expecting a changing world (e.g., Shin & Ariely, 2004).

In a longer version of this paper, I'd be excited to see
(1) a new model that addresses the mismatch between KG and human behavior;
(2) a larger-scale evaluation using both 0/1 and continuous losses, multiple horizons, and whether or not the rewards are actually static;
(3) a look at transfer learning and how performance changes across problem presentations.


--Other notes--

There were a few things that I wish had been included in the paper or done a bit differently.
* It would have been helpful to see a visualization of how well the models fit different individuals’ judgments, e.g., with error histograms.
* I wouldn't have used an ANOVA's failure-to-reject to claim there's no difference between parameter estimates across different models. If the authors think it’s important, they could summarize the inferred parameters for each model.


Lastly, I noticed a few typographical errors:
L92: “…all participants played same, but…”
L128: “…inset shows example Beta prior…”
L351: “…always the class for…”
L406: “…their ability of predicting…”
L407: “In real world…”

Summary: The paper is well-written, describes sound research, and makes incremental but meaningful contributions to important questions about reasoning and decision-making. Based on its subject matter, I expect that it would attract a small but highly-engaged audience at NIPS.
Author Feedback

Author rebuttal: We thank all reviewers for their careful reading of the manuscript and valuable comments. In the following, we address the shared concerns in "General Comments" and then individual ones as needed. We will also be sure to include all new analyses/clarifications in any revision.

General Comments

A shared concern among reviews relates to the significance/impact of the work, which we realized we could have explained more clearly up front. Previous works have suggested either optimal and complex policies or simple but suboptimal policies for human behavior in the multi-arm bandit setting. We do a systematic model comparison here, and show that human data are best explained by knowledge gradient policy that is both computationally cheap (compared to the optimal policy) and effective (better than all heuristic policies and similar to the optimal). Moreover, KG explains why subjects over explore relative to the optimal policy, and why they explore in a more targeted fashion than assumed by the heuristic policies (randomly among the non exploitative option). Concretely, we find that people adopt a learning process that (erroneously) assumes the world to be non-stationary, and that they employ a semi-myopic choice policy that is sensitive to the horizon but assumes one-step exploration when comparing action values.

Review 4

Formal measures of model comparison

We agreed with the reviewer that formal measure of model comparison is needed. We decided to do a model comparison analysis on the model agreement based on pairwise t tests (results included in the revision).

Error bars are in the caption

They are standard errors of model agreements for all tests. Specifically, we ran a 20-fold cross validation (leave one game out), resulting in 20 values of model agreement in tests. In the bar graph, we plotted each error bar as standard error of the corresponding model agreement values across tests.

Comparison with other models

We have started evaluating more policies, including a softmax policy. We will add the result in a new, journal manuscript of this research. A model of continuous change in the reward rate is definitely interesting and worthwhile, but beyond the scope of the current paper, although we plan to add some text in the Discussion related to this issue.

Reviewer 5

Model complexity

We agree that DBM is a more general model than FBM, but we think cross-validation should provide a sufficient remedy for over-fitting. Additionally, we have now used formal statistical tests to show the difference in model agreements. Please refer to our reply to reviewer 4 for the same question.

Reward structure stationarity

We think it is the fact that the subjects knew the reward structure being fixed that made the result interesting. The human brain apparently has a deep and difficult-to-overcome prior belief of a changing world. Our result is in line with many perceptual decision-making studies, where subjects tend to pick up local, transient patterns of stimuli even if they are instructed that the experiment is randomized, and past sequence of observations has no real predictive power (e.g Soetens, E, Boer, L C, & Hueting, J E 1985. JEP: HPP 11: 598-616.).

In addition, DBM does discount the past observations, making predictions based on more recent trials. However, instead of attributing this discounting behavior to an arbitrary biological constraint -- memory loss -- we alternatively interpret it as the result of the automatic engagement of mechanisms that are critical for adapting to a changing environment.

Reviewer 6

Histogram of model prediction errors

We included a figure in the supplemental materials, showing the model agreement distribution.

Related work

We appreciate that the reviewer has pointed us to two very interesting papers in human decision making in bandit problems. Bandit paradigm is in fact quite common in psychology and cognitive science literature. The two papers studied the effect on value in making options unavailable. The subjects know explicitly that the availability of option will change.

The question we posed in the current submission is quite different, and to our best knowledge, it is new. We focus on whether people do have inadvertent belief of a changing world, and we take a computational approach, finding a normative account of how people negotiate exploration and exploitation. Our analysis shows that people exhibit more exploratory behaviors than the optimal (or an approximately optimal) algorithm would do in the same environment, and our model analyses support DBM as a normative explanation of people's over-exploratory behavior: the assumption of a non-stationary world leads to extra uncertainty of the environment. In addition, KG is able to capture the decreasing trend of explorations by reducing the weight of information along the time course.

Generality of the dynamic belief model

We do believe the dynamic belief model may in general be applied to explain human behavior in many other learning and decision-making settings, not constrained on the bandit paradigm. It has successfully addressed sequential effects (i.e. inappropriate linking of stimuli or actions with consequences) in lower-order perception (for example, see Yu and Cohen in NIPS 2008). Our result suggests non-stationary world assumptions may also play a critical role in higher-order learning and decision making.

Future version

We will include softmax, and a more general version of KG where the weight of information gain becomes a psychological parameter (instead of being deterministic as in the current form). We are currently collecting our own data in multi-armed bandit, where we will be able to manipulate the horizon, reward structure, as well as the stationarity of the environment.